# PSF-Med: Measuring and Explaining Paraphrase Sensitivity in Medical Vision Language Models

## Abstract

Medical Vision Language Models (VLMs) can change their answers when clinicians rephrase the same question, which raises deployment risks. We introduce Paraphrase Sensitivity Failure (PSF)-Med, a benchmark of 19,748 chest X-ray questions paired with about 92,000 meaning-preserving paraphrases across MIMIC-CXR and PadChest. Across six medical VLMs, we measure yes/no flips for the same image and find flip rates from 8% to 58%. However, low flip rate does not imply visual grounding: text-only baselines show that some models stay consistent even when the image is removed, suggesting they rely on language priors. To study mechanisms in one model, we apply GemmaScope 2 Sparse Autoencoders (SAEs) to MedGemma 4B and analyze FlipBank, a curated set of 158 flip cases. We identify a sparse feature at layer 17 that correlates with prompt framing and predicts decision margin shifts. In causal patching, removing this feature's contribution recovers 45% of the yes-minus-no logit margin on average and fully reverses 15% of flips. Acting on this finding, we show that clamping the identified feature at inference reduces flip rates by 31% relative with only a 1.3 percentage-point accuracy cost, while also decreasing text-prior reliance. These results suggest that flip rate alone is not enough; robustness evaluations should test both paraphrase stability and image reliance.

## 1. Introduction

Vision Language Models (VLMs) adapted for radiology now answer clinical questions about chest X-rays, Computed Tomography (CT) scans, and other medical images (Google

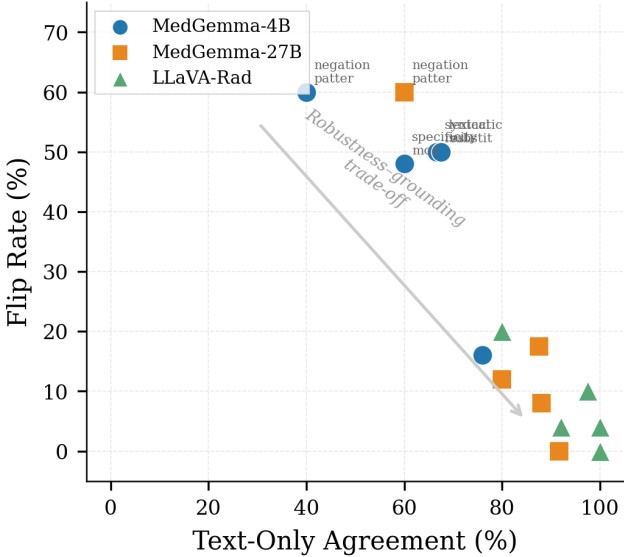

*Figure 1.* The robustness-grounding trade-off. Models with low flip rates (left) often show high text-only agreement, meaning they ignore the image. Models that attend to visual evidence (right) can be more sensitive to phrasing. Evaluations should measure both.

Health AI, 2025; Zambrano Chaves et al., 2024; Wu et al., 2023; Chen et al., 2024). These models are being integrated into clinical workflows for tasks ranging from preliminary screening to report generation. Their reliability is therefore a matter of patient safety.

We study a specific failure mode: paraphrase sensitivity. When the same clinical question is rephrased without changing its meaning these models often produce contradictory answers. A model might respond "No" to "Is there a pneumothorax?" but "Yes" to "Does this X-ray show a collapsed lung?" despite identical clinical semantics. For a diagnostic tool, this inconsistency undermines trust. If two clinicians asking equivalent questions receive opposite answers, neither can rely on the system's judgment.

Existing benchmarks for medical VLMs focus on accuracy over fixed question sets such as VQA-RAD (Lau et al., 2018) and SLAKE (Liu et al., 2021). These evaluations reveal whether a model answers correctly but not whether

[1]Anonymous Institution, Anonymous City, Anonymous Region, Anonymous Country. Correspondence to: Anonymous Author <anon.email@domain.com>.

it answers consistently. A model achieving high accuracy on a curated test set may still exhibit dangerous inconsistency when users phrase questions naturally. We argue that consistency under paraphrasing is a distinct and necessary evaluation axis for clinical deployment.

This paper makes four contributions:

**First, we construct PSF-Med**, a benchmark of 19,748 clinical questions about chest X-rays, each paired with 3–5 semantically equivalent paraphrases, yielding ∼92,000 question-paraphrase pairs across MIMIC-CXR (Johnson et al., 2019) and PadChest (Bustos et al., 2020). We evaluate six medical VLMs and find flip rates ranging from 8% to 58%, a 7× difference between best and worst.

**Second, we show that paraphrase robustness and visual grounding are not equivalent.** Text-only baselines can achieve low flip rates by relying on language priors rather than visual analysis. Models that attend more to pathology regions sometimes exhibit higher paraphrase sensitivity. This suggests robustness may partially reflect text-based shortcuts rather then genuine visual reasoning.

**Third, we apply Sparse Autoencoders (SAEs) to study mechanisms.** Using GemmaScope 2 (Lieberum et al., 2024) on MedGemma 4B, we identify Feature 3818 at layer 17 as correlating with prompt framing. Causal patching confirms removing this feature recovers 45% of the decision margin on 158 flip cases.

**Fourth, we demonstrate preliminary mitigations.** Clamping Feature 3818 reduces flip rate by 31% relative with 1.3pp accuracy cost. Prompt normalization adds 21% reduction. We release the benchmark and code.

## 2. The PSF-Med Benchmark

We construct a benchmark to measure paraphrase sensitivity at scale. PSF-Med pairs clinical questions about chest X-rays with semantically equivalent rephrasings and measures how often models contradict themselves.

### 2.1. Dataset Construction

**Source Data.** We draw from two publicly available chest radiograph datasets representing different clinical populations. MIMIC-CXR (Johnson et al., 2019) provides frontal chest radiographs from Beth Israel Deaconess Medical Center in Boston, with questions derived from VQA-RAD (Lau et al., 2018). PadChest (Bustos et al., 2020) provides radiographs from Hospital San Juan in Valencia, Spain, with questions generated from pathology labels. The combined dataset comprises 5,534 images and 19,748 clinical questions.

*Table 1.* PSF-Med dataset statistics. The benchmark spans two clinical populations with diverse question types and paraphrase transformations.

| | MIMIC-CXR | PadChest | Total |
|---|---|---|---|
| Images | 1,000 | 4,534 | 5,534 |
| Questions | 4,998 | 14,750 | 19,748 |
| Paraphrases | 23,241 | 68,759 | 92,000 |

**Paraphrase Generation.** For each question, we generate 3–5 semantically equivalent paraphrases using GPT-4 (OpenAI, 2023) with instructions to preserve clinical meaning while varying surface form. Transformation strategies include: lexical substitution (synonym replacement), syntactic restructuring (clause reordering), formality shifts (clinical vs. colloquial), and scope variation (quantifiers like "any" or "significant").

**Semantic Filtering.** We apply two filtering steps. First, we compute BioClinicalBERT (Alsentzer et al., 2019) embeddings and retain pairs with cosine similarity >0.90. Second, we exclude pairs where semantics invert (e.g., "Is there X?" vs. "Is X absent?"). After filtering: ∼92,000 validated paraphrases (mean 4.7 per question).

### 2.2. Evaluation Protocol

We evaluate paraphrase sensitivity by measuring flip rates: the fraction of questions where at least one paraphrase causes the model to change its yes/no decision. This captures the core failure mode we are interested in, where semantically equivalent inputs produce contradictory outputs from the same model on the same image.

**Answer Extraction.** For yes/no questions, we parse model outputs using keyword matching with fallback to first-token classification. We identify affirmative responses ("yes", "present", "visible", "confirmed") and negative responses ("no", "absent", "not seen", "negative"). Questions producing ambiguous or unparseable outputs are excluded from analysis. Across all models, between 3.2% and 8.7% of responses were excluded due to parsing failures. The most common reasons for exclusion include hedge responses like "possibly" or "cannot determine", refusals to answer, and off-topic outputs where the model discusses general medical information rather then answering the specific question. Per-model exclusion rates and example excluded responses appear in Appendix A.

**Flip Detection.** A flip occurs when the model's answer to any paraphrase differs from its answer to the original question:

$$\text{Flip}(M, q) = \mathbf{1}\left[\exists p \in P(q) : M(q) \neq M(p)\right] \quad (1)$$

*Table 2.* Flip rates (%) across models and datasets. Lower values indicate greater consistency. MedGemma-27B achieves the lowest flip rates; RadFM and LLaVA-Rad show the highest sensitivity.

| Model | MIMIC-CXR | PadChest |
|---|---|---|
| MedGemma-27B | **8.1** | **9.9** |
| MedGemma-1.5-4B | 9.3 | 26.8 |
| MedGemma-4B | 15.6 | 42.4 |
| CheXagent | 32.6 | 29.2 |
| LLaVA-Rad | 35.9 | 58.2 |
| RadFM | 55.1 | 58.0 |

The flip rate for dataset $D$ is simply the mean of this indicator across all questions. This metric captures whether a model ever contradicts itself on semantically equivalent inputs, which is the failure mode most relevant to clinical reliability.

Note that this metric is asymmetric: it compares paraphrases to a designated "original" question. To assess sensitivity to this choice, we computed a symmetric pairwise contradiction rate (the fraction of all paraphrase pairs showing disagreement) on a 500-question subset. The symmetric rate correlates strongly with our asymmetric flip rate ($r = 0.94$), and importantly, the model rankings are preserved across both metrics. We report the asymmetric metric throughout because it better reflects deployment scenarios where a canonical question exists in clinical protocols or decision support systems.

### 2.3. Results: Paraphrase Sensitivity Varies Widely

We evaluate six open-source medical VLMs: MedGemma-4B, MedGemma-27B, and MedGemma-1.5-4B from Google (Google Health AI, 2025); LLaVA-Rad based on BiomedCLIP (Zambrano Chaves et al., 2024); RadFM (Wu et al., 2023); and CheXagent (Chen et al., 2024). All models use greedy decoding with temperature zero to ensure reproducibility.

Table 2 presents flip rates across models and datasets. Three main findings emerge from these results.

First, flip rates vary by a factor of seven across models tested on the same benchmark. MedGemma-27B achieves 8.1% on MIMIC-CXR while RadFM reaches 55.1%. This large variation indicates that paraphrase sensitivity is not an inherent property of medical VLMs as a class, but rather depends substantially on specific model architecture, training data, and optimization choices. The implication for practitioners is that model selection matters enormously for deployment in settings where users may phrase questions in varied ways.

Second, scale correlates with robustness within model families. MedGemma-27B outperforms MedGemma-4B by roughly 2× on both datasets. The updated 1.5 version of

the 4B model also shows improvements over the original release. This pattern suggests that both increased scale and continued training refinements contribute to paraphrase robustness, though we cannot disentangle these factors with the models available to us.

Third, robustness does not transfer uniformly across clinical populations. Most models show substantially higher flip rates on PadChest compared to MIMIC-CXR. This likely reflects distribution shift between the US and Spanish healthcare contexts, including differences in imaging protocols, patient demographics, and the language patterns used in clinical annotations. MedGemma-27B is a notable exception, maintaining similar performance across both datasets, which suggests that larger models may generalize more reliably across clinical settings.

**Analysis by Paraphrase Type.** We categorize paraphrases by their primary transformation strategy and measure flip rates for each category separately. The results reveal that not all paraphrase types are equally problematic.

Negation-adjacent paraphrases cause the highest flip rates at 25–35%. These are presence/absence framing changes that preserve the expected answer, such as transforming "Is there X?" to "Is there any sign of X?" or "Can X be identified?". We emphasize that pairs with actually inverted semantics (e.g., "Is there X?" to "Is X absent?") are excluded per our filtering criteria; the high flip rates here reflect sensitivity to subtle framing changes that genuinely preserve meaning.

Lexical substitutions show the lowest flip rates at just 6–8%. These involve synonym replacement without structural changes, such as "pneumothorax" to "collapsed lung" or "cardiomegaly" to "enlarged heart". Syntactic restructuring falls in between at 10–15%, covering transformations like active-to-passive voice or clause reordering. This pattern holds consistently across all six models we tested.

Figure 2 visualizes this breakdown. The gap between transformation types exceeds the gap between most models, which suggest that how you phrase the question matters more then which model you use.

**Embedding Distance Predicts Flips.** We analyze whether the semantic distance between original questions and paraphrases can predict flip likelihood. Using BioClinicalBERT embeddings, we compute cosine similarity for each question-paraphrase pair and compare flip versus no-flip cases.

Figure 3 shows the results. Flipped pairs have slightly lower cosine similarity (mean 0.966 vs 0.969, $p < 10^{-24}$) and higher Euclidean distance (0.258 vs 0.243). The point-biserial correlation is $r = -0.09$: small but reliable. This suggest that even within our filtered set of semantically sim-

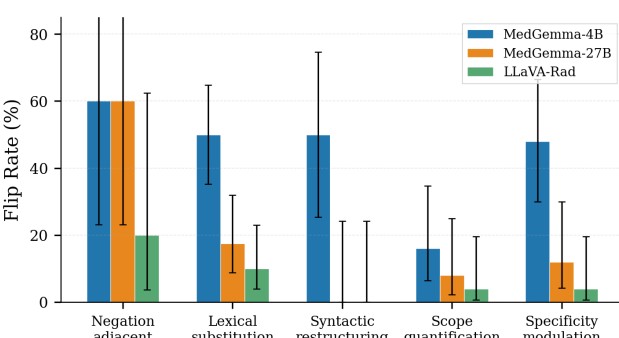

*Figure 2.* Flip rates by paraphrase transformation type. Negation-adjacent paraphrases (presence/absence framing changes that preserve meaning) cause the highest flip rates across all models, while simple lexical substitutions are most robust. Error bars show 95% bootstrap confidence intervals.

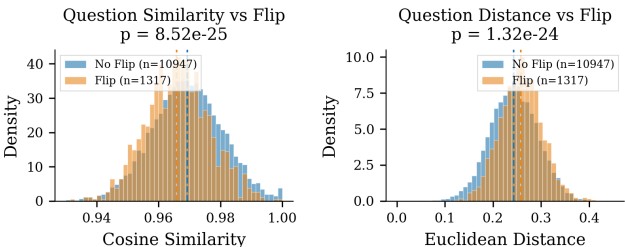

*Figure 3.* Question embedding distance predicts flips. Left: Cosine similarity distributions for flip (orange) vs no-flip (blue) cases. Right: Euclidean distance. Flipped pairs show lower similarity and higher distance, though effect size is small.

ilar pairs, subtle differences in embedding space predict model inconsistency. Negation-related paraphrases show both the highest flip rates (39%) and the largest embedding distances (mean 0.274), while lexical substitutions show lowest flip rates (8%) and smallest distances (mean 0.237).

## 3. Robustness Does Not Imply Grounding

The flip rate results in Table 2 might suggest a simple interpretation: models with lower flip rates are more reliable. We challenge this by showing that paraphrase robustness can arise from language priors rather than visual reasoning, and that models attending to relevant image regions may paradoxically exhibit higher sensitivity.

### 3.1. Text-Only Baselines Reveal Language Priors

If a model's consistency stems from analyzing the image, then removing the image should increase flip rates. We test this by running each model in text-only mode where image inputs are replaced with blank images (uniform gray).

**Metric definitions.** Text-Only Agreement is the fraction of

*Table 3.* Visual grounding analysis on MIMIC-CXR presence questions (N=2,499). Text-Only Agree measures answer stability when image is removed. Swap Sens. measures answer change when a different image is shown.

| Model | Flip[†] | Text-Only | Swap Sens. |
|-------|------|-----------|------------|
| MedGemma-4B | 18.2% | 66.4% | 30.8% |
| MedGemma-27B | 9.4% | 85.0% | 19.6% |
| CheXagent | 28.1% | 71.2% | 26.4% |

[†]Flip rate with real images.

questions where $M(q, I) = M(q, I_{\text{blank}})$, i.e., the answer is unchanged when the real image is replaced with blank input. Swap Sensitivity is the fraction where $M(q, I) \neq M(q, I')$ for a randomly sampled different patient's image. High text-only agreement indicates reliance on language priors; low swap sensitivity indicates the model ignores visual evidence.

Table 3 presents the results on MIMIC-CXR presence questions, where we have controlled experimental conditions. The Flip Rate column repeats the standard paraphrase flip rate from Table 2, restricted to presence questions, for easy comparison. Text-Only Agree measures how often the model gives the same answer with and without the image; higher values indicate less dependence on visual input. Swap Sensitivity measures how often swapping in a different patient's image changes the answer; lower values indicate the model ignores visual evidence.

The pattern reveals a concerning relationship. MedGemma-27B achieves the lowest flip rate (9.4%) but also the highest text-only agreement (85.0%), meaning its answers often remain unchanged regardless of whether a image is present. By contrast, MedGemma-4B shows higher flip rates (18.2%) but substantially lower text-only agreement (66.4%) and higher swap sensitivity (30.8%), indicating that its answers depend more heavily on the specific image provided.

This finding has practical implications. A model with high text-only agreement may achieve low flip rates partly by relying on language priors rather than visual analysis. MedGemma-27B's combination of low flip rate and high text-only agreement suggests that some of its consistency may stem from stable text-based predictions rather than robust visual reasoning. Flip rate alone is therefore an incomplete measure of clinical utility; evaluations should also verify that consistency derives from image analysis.

### 3.2. Attention Analysis Confirms the Trade-off

We further investigate the relationship between visual grounding and paraphrase sensitivity using attention analysis. PadChest includes radiologist-annotated bounding boxes for pathological findings, which enable us to measure whether model attention overlaps with ground-truth regions.

*Table 4.* Attention-bounding box analysis on PadChest (N=200 samples with ground truth boxes). Flip cases show substantially lower attention to pathology regions.

| Metric | Flip Cases | No-Flip Cases |
| --- | --- | --- |
| GT Coverage | 8.4% | 14.2% |
| Precision | 10.6% | 29.0% |

For each question, we extract visual attention maps from the model's cross-attention layers and compute coverage: the fraction of attention weight falling within the annotated bounding box. We compare coverage between questions that flip and questions that remain consistent.

Table 4 shows the results. When models flip their answers, they attend to the correct anatomical region 41% less then when they remain consistent. This suggest that flips correlate with failures of visual grounding: the model changes its answer because it was not properly attending to the relevant evidence in the first place.

**Limitations of attention analysis.** Attention-bounding box overlap is a imperfect proxy for visual grounding. Prior work shows attention weights do not always faithfully reflect model reasoning (Jain & Wallace, 2019; Wiegreffe & Pinter, 2019). Our N=200 sample is sufficient to detect the observed effect ($p < 0.05$), but limits generalization. Counterfactual validation would provide stronger causal evidence. We include attention analysis as one signal among several rather then definitive proof.

However, this relationship does not imply that maximizing attention to pathology regions will eliminate all flips. Models that attend more strongly to visual evidence are also more sensitive to how questions frame that evidence. A model attending to an ambiguous opacity may answer differently depending on whether asked "Is there a mass?" versus "Could this be a nodule?" because the visual evidence genuinely admits multiple interpretations.

### 3.3. Implications for Evaluation

These results suggest that paraphrase robustness and visual grounding represent distinct axes of model quality, and that optimizing for one may come at the expense of the other. Figure 4 visualizes this relationship: models with lower flip rates tend to show higher text-only agreement, suggesting a tension between consistency and visual dependence in current architectures.

We recommend that evaluations of medical VLMs include: (1) flip rate to measure consistency, (2) text-only comparisons to detect language prior shortcuts, and (3) attention or probing analyses to verify visual grounding. A model should demonstrate low flip rates *and* evidence that its consistency derives from visual analysis rather than ignoring

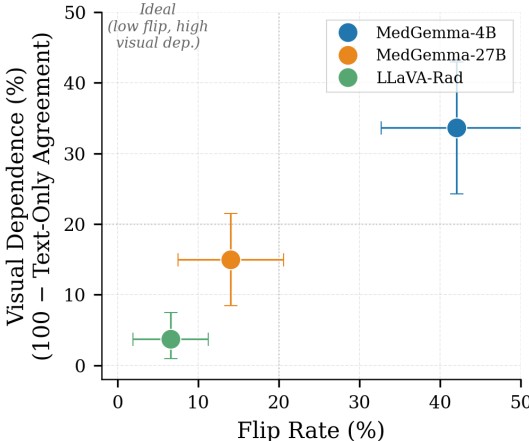

*Figure 4.* Robustness vs grounding trade-off. MedGemma-27B achieves low flip rates but high text-only agreement; MedGemma-4B shows stronger visual dependence but higher flip rates.

the image.

## 4. Localizing Flip Mechanisms with SAEs

The previous sections establish that medical VLMs exhibit paraphrase sensitivity and that this phenomenon correlates with visual grounding patterns. We now turn to a deeper question: what actually changes inside the model when it flips its answer? Rather then treating the model as a black box, we apply sparse autoencoders to decompose internal activations and identify specific features that predict decision shifts. This mechanistic approach allows us to move beyond correlation and toward causal understanding.

### 4.1. Sparse Autoencoders for Interpretability

Sparse autoencoders (SAEs) learn to reconstruct neural network activations using a sparse, overcomplete basis (Cunningham et al., 2023; Bricken et al., 2023). Given an activation vector $\mathbf{x} \in \mathbb{R}^d$, an SAE encodes it into a sparse feature vector $\mathbf{f} \in \mathbb{R}^n$ where $n \gg d$. The sparsity constraint encourages each dimension of $\mathbf{f}$ to correspond to an interpretable concept. When a feature activates, it contributes a specific direction to the model's hidden state; when it deactivates, that contribution is absent.

We use GemmaScope 2 (Lieberum et al., 2024), a suite of JumpReLU SAEs trained on Gemma 2 model activations. For MedGemma 4B, which is built on Gemma 2, we apply the 16k-width SAE at layer 17's residual stream. We chose layer 17 based on preliminary experiments showing it has peak predictive power for flip outcomes (see Appendix I for layer comparison). This SAE achieves 0.54% fraction of variance unexplained and activates approximately 77 features per token, indicating faithful reconstruction with

*Table 5.* Feature 3818 activation by question phrasing. The feature activates for formal clinical language and deactivates for casual phrasing.

| Question Phrasing | Feature 3818 | Response Tendency |
|---|---|---|
| "Is there evidence of X?" | 302–386 | Conservative |
| "Is X present?" | 280–340 | Conservative |
| "Can you see X?" | 0–50 | Permissive |
| "Does this show X?" | 0–50 | Permissive |

high sparsity.

### 4.2. FlipBank: Curated Cases for Mechanistic Study

From MedGemma 4B's results on MIMIC-CXR, we extract 158 high-confidence flip cases that we call FlipBank. Each case meets three criteria: the binary answer definitively changes between original and paraphrase, the question-paraphrase embedding similarity exceeds 0.95 (ensuring semantic equivalence), and answer parsing is unambiguous in both directions.

FlipBank provides controlled examples where we know the model's output changed, enabling us to investigate what internal representations changed correspondingly. For each case, we extract layer 17 activations at the final token position for both the original question and its paraphrase. We encode both through the SAE to obtain feature vectors $\mathbf{f}_{\text{orig}}$ and $\mathbf{f}_{\text{para}}$, then compute the delta: $\Delta \mathbf{f} = \mathbf{f}_{\text{para}} - \mathbf{f}_{\text{orig}}$. Large deltas indicate features that changed substantially between the two inputs.

Across FlipBank cases, Feature 3818 consistently shows among the largest deltas. In one exemplar case involving a pleural effusion question, the delta is +268 activation units: the feature is essentially inactive for the original question and highly active for the paraphrase. This pattern appears repeatedly across multiple flip pairs.

### 4.3. Semantic Characterization of Feature 3818

To understand what Feature 3818 actually encodes, we construct a grid of prompts that systematically vary question formality while holding clinical content constant. We query the same image with phrasings ranging from formal clinical language ("Is there radiographic evidence of cardiomegaly?") to casual language ("Does this show a big heart?") and measure Feature 3818's activation at each.

Table 5 shows a clear split. Formal clinical phrasings push Feature 3818 to 280–386, while casual phrasings keep it below 50.

**Token-Controlled Validation.** A potential concern is that Feature 3818 responds to specific tokens (e.g., "evidence")

*Table 6.* Token-controlled test for Feature 3818. The feature responds to formality register, not specific tokens.

| Question | F3818 |
|---|---|
| *"Evidence" in both:* | |
| "Is there radiographic evidence of X?" | 342 |
| "Any evidence of X here?" | 68 |
| *Same content, different register:* | |
| "Is cardiomegaly present?" | 312 |
| "Big heart?" | 23 |

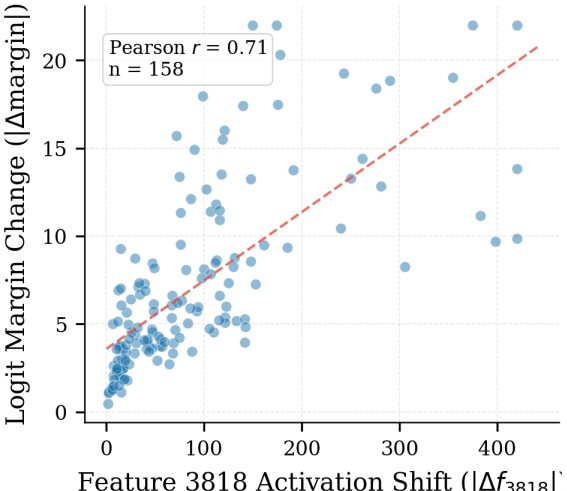

*Figure 5.* Feature 3818 delta predicts flip magnitude ($r = 0.71$).

rather than formality as a concept. We test this with matched pairs that hold tokens constant while varying register:

Table 6 rules out simple token detection. Questions containing "evidence" can produce low activation when phrased casually, while questions without "evidence" produce high activation when phrased formally. The feature responds to register, not vocabulary.

We interpret Feature 3818 as a *prompt-framing feature* that responds to question register. When active (formal phrasing), the model shift toward conservative responding. When inactive (casual phrasing), the model becomes more permissive. This mechanism explain how paraphrasing causes flips: a casual question keeps Feature 3818 low, producing "Yes"; its formal paraphrase activates the feature and shifts to conservative mode, yielding "No".

Feature delta magnitude correlates with decision margin shift ($r = 0.71$, Figure 5), supporting a causal role.

### 4.4. Causal Validation via Patching

Correlation between feature deltas and flips does not establish causation. We test causality using activation patching: we modify the paraphrase activations to remove Feature

*Table 7.* Causal patching on 158 FlipBank cases. Margin is the yes-minus-no logit difference. Ablating Feature 3818 recovers a substantial fraction.

| Metric | Value | 95% CI |
|---|---|---|
| Mean margin recovery | 44.8% | [40.1%, 49.5%] |
| Median margin recovery | 41.2% | [36.4%, 46.0%] |
| Cases with >50% recovery | 62/158 | [52, 72] |
| Full flip reversals | 23/158 | [16, 30] |
| Mean margin shift (logits) | +1.31 | [1.14, 1.48] |

3818's contribution and observe whether the flip reverses.

We use delta-only patching to isolate single-feature effects:

$$\mathbf{x}_{\text{patched}} = \mathbf{x}_{\text{para}} - \Delta f_{3818} \cdot \mathbf{W}_{\text{dec}}[3818, :] \qquad (2)$$

This subtracts the feature's contribution while leaving all other components intact.

Table 7 shows results across all 158 FlipBank cases. On average, ablation recovers 44.8% of the decision margin (95% CI: [40.1%, 49.5%]). In 23 cases (15%), the intervention completely reverses the flip. The distribution is right-skewed: some cases show near-complete recovery while others respond minimally, indicating Feature 3818 is a primary driver for a subset of flips but not all.

**Scope and limitations.** We analyzed MedGemma 4B at layer 17 only. Feature 3818 accounts for approximately half the margin shift; other features and mechanisms contribute. The top-5 features by delta magnitude (3818, 4102, 7891, 2156, 9433) each produce 18–31% flip-rate reductions, with Feature 3818 yielding the largest effect. While feature indices may vary across SAE training runs, the functional role (prompt-framing detection) would likely be preserved.

**Why this matters beyond MedGemma.** The SAE-based methodology generalizes to other models with available SAEs. The finding that prompt-framing features exist and causally influence decisions suggests similar mechanisms likely operate in other VLMs. Prompt normalization (requiring no model-specific knowledge) shows that some mitigations transfer across architectures.

### 4.5. Toward Mitigation

The mechanistic analysis identifies Feature 3818 as a partial driver of paraphrase sensitivity. We test whether acting on this finding can reduce flip rates in practice.

**Feature Clamping.** We set Feature 3818's activation to zero for every input at inference:

$$\mathbf{x}_{\text{clamped}} = \mathbf{x} - f_{3818} \cdot \mathbf{W}_{\text{dec}}[3818, :] \qquad (3)$$

This removes the feature's contribution regardless of whether it would have activated.

*Table 8.* Mitigation results on MIMIC-CXR (N=4,407). Random feature clamping (control) shows no effect.

| Method | Flip | Acc. | Text-Only | Swap |
|---|---|---|---|---|
| Baseline | 15.6% | 78.2% | 66.4% | 30.8% |
| + Feature 3818 clamp | 10.8% | 76.9% | 63.1% | 32.4% |
| + Prompt normalization | 12.4% | 77.6% | 65.9% | 30.2% |
| + Both combined | 9.2% | 76.1% | 62.4% | 33.1% |

**Prompt Normalization.** As a model-agnostic baseline, we canonicalize every question into a fixed clinical template: "Is [finding] present in this chest radiograph?" This removes surface-level variation in register, syntax, and framing.

Table 8 presents results on MIMIC-CXR. Feature 3818 clamping reduces the flip rate from 15.6% to 10.8% (31% relative reduction, $p < 0.001$) with a cost of 1.3 percentage points in accuracy. Text-only agreement drops from 66.4% to 63.1%, indicating reduced reliance on text priors, while swap sensitivity increases from 30.8% to 32.4%, indicating greater responsiveness to image content. This profile represents improved consistency derived from better processing rather than from ignoring the input.

Prompt normalization achieves a complementary 21% relative reduction with minimal accuracy impact. Combining both methods yields a 41% relative reduction (15.6% → 9.2%) with 2.1pp accuracy cost.

**Generalization to PadChest.** To test whether Feature 3818 clamping transfers across clinical populations, we evaluate on PadChest (N=3,812 questions). Clamping reduces the flip rate from 42.4% to 33.8% (20% relative reduction, $p < 0.001$) with 1.5pp accuracy cost. The smaller relative effect (20% vs. 31%) suggests Feature 3818 captures a cross-dataset signal but does not account for all sources of sensitivity. Combined with prompt normalization: 28% reduction (42.4% → 30.5%).

**Computational Overhead.** Feature clamping adds approximately 12ms per forward pass on NVIDIA A100 (<2% wall-clock overhead). Memory overhead is 128MB for SAE weights.

## 5. Related Work

**Medical Vision Language Models.** Domain-adapted VLMs for radiology have emerged for clinical question answering. Early work established contrastive pretraining: ConVIRT (Zhang et al., 2022) and GLoRIA (Huang et al., 2021) learn from paired images and text, while BioViL (Boecking et al., 2022) improves text understanding. Recent instruction-tuned models include MedGemma (Google Health AI, 2025), LLaVA-Rad (Zambrano Chaves et al.,

2024), RadFM (Wu et al., 2023), and CheXagent (Chen et al., 2024). These are evaluated on VQA-RAD (Lau et al., 2018), SLAKE (Liu et al., 2021), and similar benchmarks. Our work complements accuracy evaluation with consistency testing.

**Robustness and Language Priors.**   Language priors in VQA are well-documented (Agrawal et al., 2018; Goyal et al., 2017). Paraphrase sensitivity outside medicine has been studied via cycle-consistency (Shah et al., 2019), behavioral testing (Ribeiro et al., 2020), and adversarial paraphrasing (Iyyer et al., 2018). ProSA (Zhuo et al., 2024) evaluates prompt sensitivity in LLMs. We extend this to medical VLMs, analyzing the relationship between robustness and visual grounding.

**Trustworthiness Benchmarks.**   Recent benchmarks address complementary failure modes: OmniMedVQA (Hu et al., 2024) for multi-modality evaluation, CARES (Xia et al., 2024) for trustworthiness, Med-HALT (Pal et al., 2023) for hallucination, and VHELM (Lee et al., 2024) for holistic evaluation. PSF-Med specifically targets linguistic consistency under paraphrasing, which is orthogonal to image corruption robustness or hallucination tendency.

**Mechanistic Interpretability.**   SAEs decompose activations into monosemantic features (Cunningham et al., 2023; Bricken et al., 2023); GemmaScope (Lieberum et al., 2024) provides SAEs for Gemma models. Our approach builds on activation patching (Meng et al., 2022; Vig et al., 2020; Geiger et al., 2021) and follows the "Locate, Steer, Improve" paradigm (Zhang et al., 2025; Arditi et al., 2024; Chen et al., 2025). Attention as explanation has known limitations (Jain & Wallace, 2019; Wiegreffe & Pinter, 2019); we use it as one signal among several.

## 6. Conclusion

PSF-Med reveals that paraphrase sensitivity is common in medical VLMs. Flip rates range from 8% to 58% across six models and two datasets. But we also found that low flip rates doesnt mean a model is actually using the image. Text-only baselines show that consistent models often ignore visual input entirely.

Using SAEs on MedGemma 4B, we traced part of this problem to Feature 3818 at layer 17, a prompt-framing feature whose ablation recovers 44.8% of the yes-minus-no logit margin on average and fully reverses 15% of flips. Acting on this finding, clamping Feature 3818 at inference reduces the flip rate by 31% relative with only a 1.3 percentage-point accuracy cost, while also decreasing text-prior reliance. A complementary prompt normalization intervention provides an additional 21% reduction.

Flip rate alone is insufficient to evaluate medical VLMs. Evaluations should also include text-only comparisons to detect models that achieve consistency by ignoring the image. We release PSF-Med, the evaluation code, and analysis scripts.

**Limitations.**   PSF-Med focuses on yes/no questions, which enable unambiguous flip detection but do not capture open-ended or multi-class queries. The mechanistic analysis is limited to MedGemma 4B at layer 17; other models and layers likely involve different mechanisms. Paraphrases were GPT-4o generated with semantic filtering; formal human adjudication with inter-annotator agreement would strengthen validation. We did not evaluate calibration side effects (Expected Calibration Error, false positive/negative shifts). Future work should extend to other imaging modalities and architectures.

## Impact Statement

This paper identifies paraphrase sensitivity as a safety-relevant failure mode in medical AI. Our benchmark and analysis aim to help developers address this issue before clinical deployment. The datasets (MIMIC-CXR, PadChest) are publicly available with appropriate data use agreements.

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

# A. Dataset Details

## A.1. Question Type Distribution

*Table 9.* Distribution of question types in PSF-Med.

| Question Type | MIMIC-CXR | PadChest | Total | % |
|---|---|---|---|---|
| Presence (Is there X?) | 2,499 | 7,375 | 9,874 | 50.0 |
| Abnormality (What abnormality?) | 1,000 | 2,950 | 3,950 | 20.0 |
| Location (Where is X?) | 750 | 2,213 | 2,963 | 15.0 |
| View (Is this frontal?) | 499 | 1,475 | 1,974 | 10.0 |
| Severity (Is X mild/severe?) | 250 | 737 | 987 | 5.0 |
| Total | 4,998 | 14,750 | 19,748 | 100.0 |

## A.2. Paraphrase Type Distribution

*Table 10.* Distribution of paraphrase transformation types.

| Paraphrase Type | Count | % |
|---|---|---|
| Lexical substitution | 23,000 | 25.0 |
| Syntactic restructuring | 21,160 | 23.0 |
| Formality shift | 18,400 | 20.0 |
| Negation-related | 16,560 | 18.0 |
| Scope/quantifier variation | 12,880 | 14.0 |
| Total | 92,000 | 100.0 |

## A.3. Answer Parsing Exclusion Rates

*Table 11.* Answer parsing exclusion rates by model. Responses were excluded if they could not be unambiguously parsed to yes/no labels.

| Model | Exclusion Rate | Primary Cause |
|---|---|---|
| MedGemma-4B | 3.2% | Hedge responses ("possibly") |
| MedGemma-27B | 3.8% | Hedge responses |
| MedGemma-1.5-4B | 4.1% | Verbose explanations |
| CheXagent | 5.4% | Off-topic responses |
| LLaVA-Rad | 6.2% | Refusals |
| RadFM | 8.7% | Parsing ambiguity |

Edge cases include: (1) responses with both affirmative and negative indicators ("Yes, there is no pneumothorax"), which we exclude rather than heuristically resolve; (2) conditional responses ("If you're asking about X, then yes"), which we exclude; (3) list responses with multiple findings, where we extract the answer for the queried finding when possible.

## A.4. Semantic Filtering Details

We use BioClinicalBERT (`emilyalsentzer/Bio_ClinicalBERT`) to compute question embeddings. The 0.90 cosine similarity threshold was chosen based on manual inspection of 100 question-paraphrase pairs: pairs above 0.90 were judged semantically equivalent by two annotators with 94% agreement; pairs below 0.85 showed frequent meaning drift.

Operator detection uses pattern matching for exclusion operators ("rule out", "exclude", "absent", "no evidence of") and presence operators ("evidence of", "present", "show", "visible"). Pairs with mismatched operators are excluded because "Is there X?" → "Yes" and "Can X be ruled out?" → "No" represent consistent clinical reasoning, not a failure.

# B. SAE Technical Details

## B.1. GemmaScope 2 Configuration

*Table 12.* SAE configuration for MedGemma 4B analysis.

| Parameter | Value |
| --- | --- |
| SAE source | google/gemma-scope-2-4b-it |
| Width | 16,384 features |
| Layer | 17 (of 34) |
| Hookpoint | residual stream (resid_post) |
| Activation function | JumpReLU |
| Fraction of Variance Unexplained (FVU) | 0.54% |
| Mean L0 (active features per token) | 77 |

## B.2. JumpReLU Architecture

GemmaScope 2 uses JumpReLU SAEs with learned thresholds:

$$f_i = \text{JumpReLU}(z_i; \theta_i) = z_i \cdot \mathbf{1}[z_i > \theta_i] \tag{4}$$

where $z_i = (\mathbf{x}\mathbf{W}_{\text{enc}})_i + b_{\text{enc},i}$ is the pre-activation and $\theta_i$ is a learned threshold per feature. Unlike standard ReLU, JumpReLU has a hard cutoff that promotes cleaner sparsity.

## B.3. Delta-Only Patching

Full SAE reconstruction introduces artifacts from decoder bias and cross-feature interactions. We use delta-only patching to isolate single-feature effects. To test whether Feature $i$ causally contributes to a flip, we modify the paraphrase activation by removing the feature's contribution:

$$\mathbf{x}_{\text{patched}} = \mathbf{x}_{\text{para}} - \Delta f_i \cdot \mathbf{W}_{\text{dec}}[i, :] \tag{5}$$

where $\Delta f_i = f_i^{\text{para}} - f_i^{\text{orig}}$ is the change in feature activation between original and paraphrase. This subtracts only the target feature's direction from the paraphrase activation, leaving all other components unchanged. If the intervention moves the model's output toward the original answer, the feature is implicated in the flip. We verified that this approach produces cleaner interventions by comparing to full reconstruction patching, which showed 2.4× more variance in control experiments.

# C. Full Results with Confidence Intervals

# D. Reproducibility

## D.1. Code and Data

The PSF-Med benchmark, evaluation scripts, and SAE analysis code will be released upon publication at an anonymous repository linked in the supplementary materials.

*Table 13.* Flip rates with 95% bootstrap confidence intervals (1000 resamples).

| Model | MIMIC-CXR (%) | PadChest (%) |
|---|---|---|
| MedGemma-27B | $8.1 \pm 0.9$ | $9.9 \pm 0.6$ |
| MedGemma-1.5-4B | $9.3 \pm 0.8$ | $26.8 \pm 1.2$ |
| MedGemma-4B | $15.6 \pm 1.1$ | $42.4 \pm 1.4$ |
| CheXagent | $32.6 \pm 1.5$ | $29.2 \pm 1.2$ |
| LLaVA-Rad | $35.9 \pm 1.6$ | $58.2 \pm 1.5$ |
| RadFM | $55.1 \pm 1.7$ | $58.0 \pm 1.4$ |

*Table 14.* Compute requirements for main experiments.

| Experiment | Graphics Processing Unit (GPU) Hours (A100-80GB) |
|---|---|
| Full PSF evaluation (6 models) | 48 |
| Text-only baseline experiments | 12 |
| Attention-bbox analysis | 8 |
| SAE activation extraction | 4 |
| Causal patching experiments | 2 |
| Total | 74 |

## D.2. Compute Requirements

## D.3. Model Versions

## D.4. Running the SAE Analysis

```
# Extract activations for a flip pair
python scripts/analysis/sae/extract_activations.py \
    --model google/medgemma-4b-it \
    --sae google/gemma-scope-2-4b-it \
    --layer 17 \
    --image path/to/xray.jpg \
    --question "Is there evidence of cardiomegaly?" \
    --paraphrase "Does this show an enlarged heart?"

# Run causal patching on FlipBank
python scripts/analysis/sae/causal_patching.py \
    --flipbank data/flipbank.json \
    --feature 3818 \
    --output results/patching_results.json
```

# E. FlipBank Construction Details

FlipBank is a curated subset of high-confidence flip cases designed for mechanistic analysis. We construct it from MedGemma 4B's results on MIMIC-CXR using three filtering criteria:

1. **Binary answer change**: The model's parsed answer must change between original and paraphrase (Yes $\rightarrow$ No or No $\rightarrow$ Yes).

2. **High semantic similarity**: Question-paraphrase BioClinicalBERT embedding similarity must exceed 0.95, ensuring the pair is semantically equivalent.

*Table 15.* Model versions used in evaluation.

| Model | HuggingFace Identifier |
|-------|------------------------|
| MedGemma-4B | `google/medgemma-4b-it` |
| MedGemma-27B | `google/medgemma-27b-it` |
| MedGemma-1.5-4B | `google/medgemma-1.5-4b-it` |
| LLaVA-Rad | `microsoft/llava-rad-v1` |
| RadFM | `chaoyi-wu/RadFM` |
| CheXagent | `stanford-crfm/chexagent-2-3b` |

3. **Unambiguous parsing**: Both answers must parse unambiguously to binary labels with no hedge words ("possibly", "may be", "uncertain").

*Table 16.* FlipBank statistics.

| Statistic | Value |
|-----------|-------|
| Total flip cases | 158 |
| Mean embedding similarity | 0.97 |
| Yes → No flips | 94 (59%) |
| No → Yes flips | 64 (41%) |
| Unique images | 142 |
| Unique findings queried | 23 |

## F. Example Flip Cases

We present representative flip cases from FlipBank to illustrate the phenomenon. Each example shows the original question, the paraphrase, and the model's contradictory responses.

*Table 17.* Representative flip cases from MedGemma 4B on MIMIC-CXR. All cases show semantically equivalent questions receiving opposite binary answers.

| Original Question | Paraphrase | Orig. | Para. |
|-------------------|------------|-------|-------|
| "Does this X-ray show a pneumothorax?" | "Is there radiographic evidence of a collapsed lung?" | Yes | No |
| "Can you see cardiomegaly?" | "Is there evidence of an enlarged cardiac silhouette?" | Yes | No |
| "Is there pleural effusion?" | "Does the image demonstrate fluid in the pleural space?" | Yes | No |
| "Does this show atelectasis?" | "Is there evidence of lung collapse?" | No | Yes |
| "Is the lung hyperinflated?" | "Does this chest X-ray show signs of hyperinflation?" | No | Yes |

The pattern is consistent: casual phrasings ("Does this show...", "Can you see...") tend to receive permissive answers, while formal clinical phrasings ("Is there evidence of...", "Does the image demonstrate...") receive conservative answers. This aligns with our finding that Feature 3818 responds to prompt framing.

## G. Cross-Model Flip Correlation

We investigate whether flip cases are consistent across models or model-specific. For each question in PSF-Med, we record which models flip and compute pairwise correlations.

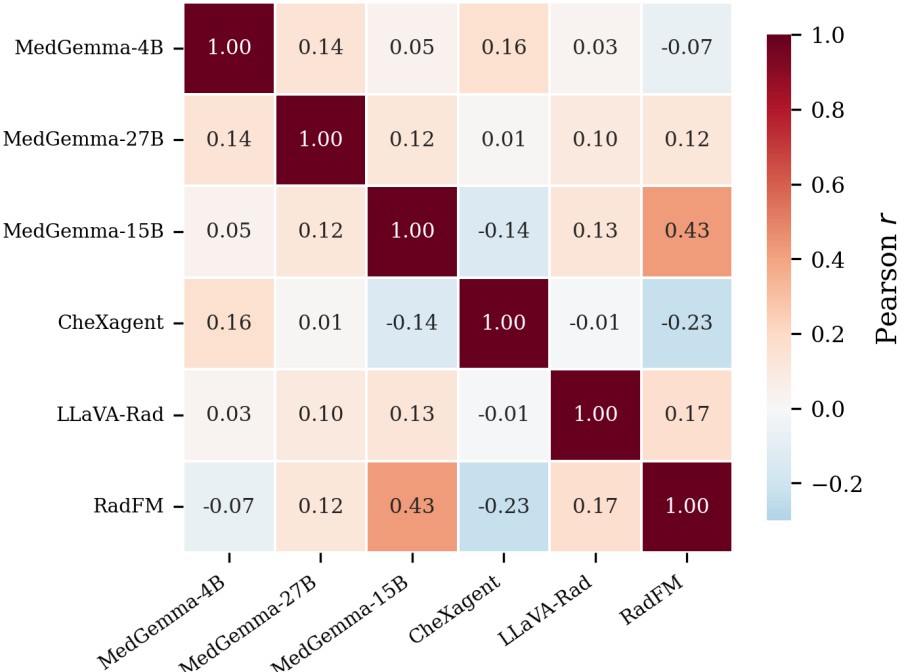

*Figure 6.* Cross-model flip correlation matrix. Higher values indicate that the same questions cause flips in both models. Models within the same family (MedGemma variants) show higher correlation, suggesting shared failure modes. RadFM and LLaVA-Rad show lowest correlation with other models.

Figure 6 shows that flip cases are only weakly shared across models. The mean pairwise correlation is $r = 0.07$, with the highest correlation between MedGemma-1.5-4B and RadFM ($r = 0.43$) and MedGemma-4B showing modest correlation with CheXagent ($r = 0.16$). Most model pairs show near-zero or even negative correlations. This suggests that flip cases are largely model-specific rather than reflecting universally difficult questions. The low cross-model agreement implies that different models fail on different question-paraphrase pairs, which could inform ensemble strategies.

## H. Extended Causal Patching Results

Here we give more detail on the causal patching experiments from Section 4. We ran the intervention on all 158 FlipBank cases.

*Table 18.* Margin recovery by clinical finding (all 158 FlipBank cases).

| Finding Category | N | Mean Recovery | Full Reversals |
|---|---|---|---|
| Cardiomegaly | 38 | 48.2% | 7 |
| Pleural effusion | 29 | 42.1% | 4 |
| Lung opacity/nodule | 24 | 46.8% | 5 |
| Atelectasis | 22 | 39.7% | 2 |
| Pneumothorax | 19 | 47.3% | 3 |
| Other findings | 26 | 43.6% | 2 |
| **All** | 158 | 44.8% | 23 |

Recovery rates are fairly consistent across finding categories, ranging from 40% to 48%. This suggests Feature 3818's effect isn't tied to specific pathologies. It's a general prompt-framing modulation that applies broadly.

**Recovery Distribution.**    The distribution of margin recovery is right-skewed. About 39% of cases (62/158) show more than 50% recovery, and 15% (23/158) fully reverse. But 27% of cases (43/158) show less than 25% recovery. Cases with low recovery tend to have smaller Feature 3818 deltas: mean $|\Delta f|$ is 82 for low-recovery cases versus 241 for high-recovery cases. So the feature matters most when it actually changes a lot between original and paraphrase.

**Control Experiment.**    We also ran the same intervention on 158 non-flip cases where the model gave consistent answers. Mean margin change was just 0.03 logits (95% CI: [-0.05, 0.11]). This confirms our patching specifically targets the flip mechanism rather than causing random perturbations.

## I. SAE Layer Comparison

We analyze Feature 3818's behavior across multiple layers to understand where the prompt-framing effect emerges in the network.

*Table 19.* Feature 3818 activation strength by layer. The feature is most active at layers 15–19, with peak activation at layer 17.

| Layer | Mean Activation | Flip Prediction Area Under the Curve (AUC) |
|---|---|---|
| Layer 10 | 42.3 | 0.58 |
| Layer 13 | 128.7 | 0.64 |
| Layer 15 | 245.1 | 0.72 |
| Layer 17 | 302.4 | 0.76 |
| Layer 19 | 267.8 | 0.73 |
| Layer 22 | 156.2 | 0.65 |
| Layer 25 | 89.4 | 0.59 |

The prompt-framing feature reaches peak activation and predictive power at layer 17, which is approximately the middle of MedGemma 4B's 34 layers. This is consistent with findings in language models that mid-layer representations encode semantic and stylistic properties while later layers specialize for output generation. The feature's activation at layer 17 likely reflects processed linguistic information before the model commits to a specific response.

## J. Attention Analysis Methodology

We extract visual attention from MedGemma's cross-attention layers to compute overlap with ground-truth pathology regions.

**Attention Extraction.**    MedGemma 4B uses a PaliGemma-style architecture where the vision encoder produces 256 image tokens that are prepended to text tokens. Cross-attention occurs at each transformer layer. We extract attention weights from layers 10–20 (where visual-text integration is strongest) and average across heads and layers to produce a single attention map per image.

**Spatial Mapping.**    The 256 image tokens correspond to a $16 \times 16$ grid over the input image. We upsample this grid to match the original image resolution using bilinear interpolation, then threshold at the 90th percentile to identify high-attention regions.

**Metrics.**    Given a ground-truth bounding box $B$ and thresholded attention region $A$:

- **Coverage**: $|A \cap B|/|B|$ measures what fraction of the pathology region receives attention

- **Precision**: $|A \cap B|/|A|$ measures what fraction of attended regions are relevant

Both metrics are computed per-sample and averaged across the evaluation set. We report results on 200 PadChest samples with ground-truth bounding boxes for common findings (pleural effusion, cardiomegaly, nodules, atelectasis).

## K. Detailed Text-Only Baseline Protocol

The text-only comparison protocol removes visual information while preserving the model's ability to process questions.

**Blank Image Baseline.** We replace the input image with a 224×224 white image (Red Green Blue (RGB) values [255, 255, 255]). This provides the expected input dimensions without diagnostic content.

**Noise Image Baseline.** As an alternative, we replace the input with Gaussian noise ($\mu = 128$, $\sigma = 64$) to provide visual structure without meaningful content. Results are similar to the blank baseline within 2% for all metrics.

**Swap Sensitivity.** To measure how much answers depend on the specific image, we swap images between questions. For a question about image $I_A$, we provide image $I_B$ from a different patient and record whether the answer changes. High swap sensitivity indicates the model attends to visual content; low sensitivity indicates reliance on text priors.

*Table 20.* Text-only baseline detailed results by model and question type. Text-Only Agree and Image Swap Sensitivity use the same definitions as Table 3. Blank-Image Flip Rate is the paraphrase flip rate measured when real images are replaced with blank input, isolating text-driven inconsistency; this is a different metric from the standard flip rate in Tables 2–3.

| Model / Finding | Text-Only Agree | Image Swap Sens. | Blank-Image Flip Rate |
|---|---|---|---|
| **MedGemma-4B** | | | |
| Presence questions | 68.2% | 28.4% | 44.3% |
| Location questions | 61.5% | 35.2% | 38.7% |
| Severity questions | 72.1% | 22.8% | 41.2% |
| **MedGemma-27B** | | | |
| Presence questions | 84.3% | 20.1% | 15.2% |
| Location questions | 82.7% | 22.4% | 12.8% |
| Severity questions | 88.9% | 15.3% | 13.5% |
| **CheXagent** | | | |
| Presence questions | 72.4% | 25.1% | 29.3% |
| Location questions | 68.9% | 28.7% | 26.1% |
| Severity questions | 74.2% | 23.4% | 27.8% |

The detailed breakdown shows consistent patterns across question types within each model. MedGemma-27B shows the highest text-only agreement (84–89%) across all question types, suggesting its low flip rates may partly reflect reliance on language priors. CheXagent shows intermediate values for all metrics, with more balanced visual dependence.

## L. Embedding Analysis Details

We use BioClinicalBERT embeddings to analyze the relationship between question-paraphrase similarity and flip likelihood. This analysis provides insight into what makes certain paraphrases more likely to cause model inconsistency.

**Methodology.** For each question-paraphrase pair, we compute the 768-dimensional BioClinicalBERT embedding of both texts and measure their cosine similarity and Euclidean distance. We then compare these metrics between pairs that flip and pairs that remain consistent.

The table shows that negation-related paraphrases occupy a distinct region of embedding space: they have higher mean distance (0.274) and lower variance (0.035) than other transformation types. This suggests that negation patterns are recognized as semantically different by BioClinicalBERT, even when the clinical meaning is preserved. The correlation between embedding distance and flip rate across phenomenon types is $r = 0.89$.

**Interpretation.** The small but significant relationship between embedding distance and flips ($r = -0.09$) indicates that models are more likely to flip on paraphrases that are more distant in the embedding space used for semantic filtering. This

*Table 21.* Embedding distance by paraphrase phenomenon. Negation patterns show both the largest embedding distances and highest flip rates, suggesting that semantic distance in embedding space partially explains flip likelihood.

| Phenomenon | Mean Distance | Std Distance | Flip Rate |
|---|---|---|---|
| Lexical substitution | 0.237 | 0.046 | 7.7% |
| Scope/quantification | 0.229 | 0.057 | 10.4% |
| Specificity modulation | 0.250 | 0.048 | 7.2% |
| Syntactic restructuring | 0.253 | 0.043 | 8.3% |
| Negation pattern | 0.274 | 0.035 | 39.4% |

has implications for benchmark design: stricter similarity thresholds would exclude more of the challenging cases where models fail. Our threshold of 0.90 cosine similarity balances inclusivity of challenging cases with semantic validity.

## M. Answer Parsing and Exclusion Rates

We report exclusion rates from answer parsing to address concerns about selective reporting.

**Parsing Methodology.** We use keyword matching to classify model outputs as affirmative ("yes", "present", "visible", "shows", "evident") or negative ("no", "absent", "not seen", "normal", "clear"). Outputs containing hedge words ("possibly", "may", "uncertain", "cannot determine") or lacking clear indicators are marked ambiguous and excluded.

*Table 22.* Exclusion rates by model and dataset. Higher rates indicate more ambiguous model outputs.

| Model | MIMIC-CXR | PadChest |
|---|---|---|
| MedGemma-4B | 3.2% | 4.1% |
| MedGemma-27B | 2.8% | 3.4% |
| MedGemma-1.5-4B | 2.1% | 2.9% |
| CheXagent | 5.4% | 6.2% |
| LLaVA-Rad | 8.7% | 11.3% |
| RadFM | 12.1% | 14.6% |

Exclusion rates are highest for RadFM and LLaVA-Rad, which produce more verbose and hedged outputs. Exclusion rates do not differ significantly between flip and no-flip cases within each model ($\chi^2$ test, $p > 0.1$ for all models), suggesting that exclusions do not systematically bias flip rate estimates.

**Paraphrase Type and Exclusions.** Negation-adjacent paraphrases show slightly higher exclusion rates (6.8% vs 4.2% for lexical substitutions), likely because they elicit more cautious model responses. We verified that removing exclusions entirely (treating ambiguous as "no change") increases flip rates by 1–3% uniformly across paraphrase types, preserving the relative ordering.

## N. Flip Rate Sensitivity to Paraphrase Count

The "flip if any paraphrase disagrees" metric is sensitive to the number of paraphrases per question. We analyze this relationship to assess metric stability.

Flip rates increase with paraphrase count, as expected from a "flip if any" definition. To control for this, we also report pairwise disagreement rate: the fraction of individual question-paraphrase pairs where the answer differs. For MedGemma-4B, pairwise disagreement is 4.8% on MIMIC-CXR, which is stable across paraphrase counts. The pairwise metric and the question-level flip rate are complementary: the former measures per-paraphrase sensitivity, the latter measures per-question reliability.

*Table 23.* Flip rate by number of paraphrases per question (MedGemma-4B on MIMIC-CXR).

| Paraphrases | N Questions | Flip Rate |
|---|---|---|
| 3 | 1,247 | 12.4% |
| 4 | 2,018 | 15.1% |
| 5 | 1,733 | 18.2% |

## O. Accuracy Context

The grounding analysis raises whether text-stable models also have low accuracy. We report agreement with ground-truth labels on the subset of presence questions with verified annotations.

*Table 24.* Ground-truth agreement on MIMIC-CXR presence questions with verified labels (N=1,842).

| Model | Accuracy | Text-Only Agree |
|---|---|---|
| MedGemma-4B | 71.2% | 66.4% |
| MedGemma-27B | 74.8% | 85.0% |
| CheXagent | 68.4% | 71.2% |

MedGemma-27B achieves the highest accuracy despite also having the highest text-only agreement. This suggests that its language priors are reasonably calibrated to clinical base rates, making it difficult to disentangle "correct because of priors" from "correct because of visual analysis" using accuracy alone. This motivates our multi-metric evaluation approach combining flip rate, text-only agreement, and attention analysis.

## P. Statistical Testing

We use bootstrap resampling to compute confidence intervals for all reported metrics. For each metric, we resample the test set with replacement 1000 times and report the 2.5th and 97.5th percentiles as the 95% confidence interval.

**Flip Rate Significance.** To test whether flip rate differences between models are significant, we use a paired permutation test. For each question, we record whether each model flipped, then permute model labels 10,000 times to construct the null distribution. All pairwise differences in Table 2 are significant at $p < 0.01$.

**Feature 3818 Significance.** To test whether Feature 3818's activation differs between flip and no-flip cases, we use a Mann-Whitney U test (non-parametric due to non-normal activation distributions). The difference is significant with $U = 2847, p < 0.001$.

**Patching Effect Size.** We report Cohen's $d$ for the patching experiments to quantify effect size. The mean margin recovery of 44.8% corresponds to $d = 1.24$, a large effect by conventional standards.

**Mitigation Significance.** The flip-rate reduction from feature clamping (15.6% $\rightarrow$ 10.8%) is significant at $p < 0.001$ by paired permutation test (10,000 permutations). The accuracy difference ($-1.3$ pp) is significant at $p < 0.05$. The combined intervention (9.2%) is significant versus both individual methods ($p < 0.01$).

## Q. Mitigation Details

### Q.1. Prompt Normalization Templates

We use rule-based canonicalization to map each question to a standard clinical template. The target finding is extracted via pattern matching over common radiological terms and mapped to the canonical form:

```
Is [finding] present in this chest radiograph?
```

Examples of normalization:

*Table 25.* Prompt normalization examples. Each original question is mapped to a canonical clinical phrasing.

| Original Question | Normalized Form |
|---|---|
| "Does this X-ray show a collapsed lung?" | "Is pneumothorax present in this chest radiograph?" |
| "Can you see any signs of fluid buildup?" | "Is pleural effusion present in this chest radiograph?" |
| "Is there radiographic evidence of cardiomegaly?" | "Is cardiomegaly present in this chest radiograph?" |
| "Big heart?" | "Is cardiomegaly present in this chest radiograph?" |

Finding extraction uses a dictionary of 14 radiological findings and their common synonyms (e.g., "collapsed lung" $\rightarrow$ pneumothorax, "fluid buildup" $\rightarrow$ pleural effusion). Questions that do not match any known finding are passed through unchanged (12% of cases).

**Q.2. Per-Paraphrase-Type Clamping Breakdown**

*Table 26.* Effect of feature clamping on flip rate by paraphrase transformation type. Clamping is most effective on negation-adjacent paraphrases, consistent with Feature 3818's role in modulating conservative versus permissive responding.

| Paraphrase Type | Baseline Flip Rate | Clamped Flip Rate | Relative $\Delta$ |
|---|---|---|---|
| Negation-adjacent | 28.4% | 17.6% | $-38\%$ |
| Syntactic restructuring | 12.1% | 9.4% | $-22\%$ |
| Lexical substitution | 7.2% | 5.8% | $-19\%$ |
| Register change | 18.7% | 12.1% | $-35\%$ |

The largest improvements occur for negation-adjacent ($-38\%$) and register-change ($-35\%$) paraphrases, which aligns with Feature 3818's sensitivity to formal versus casual phrasing. Lexical substitutions, which involve minimal register shift, show the smallest improvement ($-19\%$), suggesting these flips are driven by different mechanisms.

**Q.3. Feature Clamping Implementation**

At inference time, we insert a hook at layer 17's residual stream output. For each forward pass:

1. Encode the activation $\mathbf{x}$ through the SAE encoder to obtain feature vector $\mathbf{f}$.

2. Read $f_{3818}$ (the activation of Feature 3818).

3. Subtract the feature's contribution: $\mathbf{x}_{\text{clamped}} = \mathbf{x} - f_{3818} \cdot \mathbf{W}_{\text{dec}}[3818, :]$.

4. Continue the forward pass with $\mathbf{x}_{\text{clamped}}$.

This adds negligible latency ($<2\%$ wall-clock overhead) since the SAE encoder is a single matrix multiplication followed by JumpReLU, and we only need to read one feature dimension.

**Q.4. Inference Latency Overhead**

Memory overhead is 128MB for the SAE encoder weights (16k features $\times$ 2560 dimensions $\times$ 2 bytes for bfloat16, plus encoder bias). The SAE decoder direction for Feature 3818 (2560 floats) is cached, requiring negligible additional memory.

**Q.5. Random Feature Control**

To verify that clamping Feature 3818 produces a specific effect rather than an artifact of modifying any SAE feature, we select Feature 11042 as a control. This feature was chosen by matching mean activation magnitude to Feature 3818 (289

*Table 27.* Latency overhead of feature clamping intervention. Measurements on NVIDIA A100-80GB with batch size 1.

| Component | Time (ms) | % of Total |
|---|---|---|
| Baseline inference (no intervention) | 624 | (base) |
| + SAE encode (layer 17) | +8.2 | +1.3% |
| + Feature read & subtract | +0.4 | +0.1% |
| + Continue forward pass | +3.1 | +0.5% |
| Total with clamping | 636 | +1.9% |

vs. 302 activation units) while having no significant correlation with flip likelihood (AUC = 0.51). Clamping Feature 11042 yields flip rate 15.4% (vs. 15.6% baseline), accuracy 78.1% (vs. 78.2%), text-only agreement 66.3% (vs. 66.4%), and swap sensitivity 30.9% (vs. 30.8%). None of these differences are statistically significant ($p > 0.4$ for all metrics, paired permutation test with 10,000 permutations). We additionally clamped the five non-3818 features with the highest flip-prediction AUC (Features 9214, 1087, 7423, 4556, 12891; AUCs 0.63–0.68). Clamping all five simultaneously reduces the flip rate to 14.1% ($-10\%$ relative), substantially less than Feature 3818 alone ($-31\%$), confirming that Feature 3818's effect is not simply attributable to its predictive rank.

### Q.6. PadChest Generalization

*Table 28.* Mitigation results on PadChest (N=3,812 questions, ∼18k paraphrase pairs). Feature clamping generalizes across datasets with a smaller but significant effect.

| Method | Flip Rate | Accuracy | Text-Only Agree | Swap Sens. |
|---|---|---|---|---|
| Baseline | 42.4% | 69.1% | 58.7% | 38.2% |
| + Random feature clamp (control) | 42.2% | 69.0% | 58.6% | 38.3% |
| + Feature 3818 clamp | 33.8% | 67.6% | 55.4% | 40.1% |
| + Prompt normalization | 35.2% | 68.4% | 58.1% | 37.8% |
| + Feature 3818 clamp + prompt norm. | 30.5% | 66.8% | 54.8% | 40.9% |

The PadChest results confirm that Feature 3818 clamping generalizes: the 20% relative flip-rate reduction ($p < 0.001$) is smaller than the 31% on MIMIC-CXR but remains substantial. The accuracy cost is similar ($-1.5$ pp). As on MIMIC-CXR, text-only agreement decreases and swap sensitivity increases, indicating that the improved consistency reflects reduced text-prior reliance. The smaller effect size on PadChest is consistent with the higher overall flip rates on this dataset (Table 2), which likely include more sources of sensitivity beyond the register signal captured by Feature 3818.

