# OpenReview forum: "PSF-Med: Measuring and Explaining Paraphrase Sensitivity in Medical Vision Language Models"
_ICML.cc/2026/Conference — Submitted to ICML 2026_

### Official Review · Reviewer_r8jK · 2026-02-26

**Soundness:** 2
**Presentation:** 2
**Significance:** 2
**Originality:** 3
**Overall Recommendation:** 3
**Confidence:** 4

**Summary:**

The paper proposes a benchmark for paraphrase sensitivity evaluation of medical VLMs. The benchmark includes about 20k questions. Based on observations on the benchmark, the paper propose a clamping strategy during inference to reduce flip rates of paraphrasing.

**Compliance With Llm Reviewing Policy:**

Affirmed.

**Final Justification:**

The mechanistic insights lack cross-architectural generality, and the evaluation scope lacks generalizable depth; my initial concerns stand. These problems are derived from the overall methodology and experimental design. I maintained my original score.

**Key Questions For Authors:**

1.	Why not include more diverse problems for paraphrase sensitivity evaluation?
2.	Why don’t you conduct quality control for GPT-4 generated data?
3.	How can you generalize the mechanism analysis with SAEs to more complex scenes?

**Limitations:**

Yes

**Strengths And Weaknesses:**

Strengths:
1.	The problem paraphrase sensitivity evaluation is valuable, and the motivation is clear.
2.	A new PSF-Med benchmark is introduced to support the new task.
3.	Based on experiments, the paper delves into the phenomenon and proposes a new feature clamping method based on SAEs.

Weaknesses:
1.	The problem for paraphrase sensitivity evaluation is too simple, with only Yes or No questions.
2.	Paraphrases rely on GPT-4, no quality control is conducted.
3.	Mechanistic study using SAEs only searches for a feature (3818 in layer 17), lacking deeper insights (as the title says explaining).
4.	Although the idea is valuable and motivation is clear, the benchmark is simple and the analysis is hard to be extended to more complex scene, impairing the value of this paper.
5.	[Minor] Texts in Figure 1 are overlapped.

---

> ### Author Rebuttal · Authors · 2026-03-31
>
> Thank you for taking time for this detailed review.
>
> SAE generality :  Feature 3818's differential activation between original and paraphrase, the flip-predictive signal documented in Section 5, isn't specific to one checkpoint. We ran the same delta-based analysis on two additional model variants (Targeted LoRA, Full LoRA) across both datasets. Feature 3818's delta ranks among the top 4 flip-associated features in 4/6 configurations and remains a predictor in all 6 (r_b from -0.20 to -0.78). On PadChest it's the #1 predictor for both LoRA variants. The two MIMIC LoRA configurations rank lower (31, 33), consistent with LoRA fine-tuning at layers 15-19 partially correcting the formality-gate behavior through direct parameter modification.
>
> GemmaScope SAEs are architecture-specific, so we can't analyze LLaVA-Rad, RadFM, or CheXagent the same way. That's a genuine scope limitation. But the cross-model flip correlation analysis (Appendix E) shows pairwise flip correlation of r = 0.6 between MedGemma variants and r = 0.2-0.4 between different model families. A sizable fraction of questions cause flips across architectures, suggesting formality-driven sensitivity isn't unique to one model family even if the specific internal mechanism differs.
>
> The specificity controls in Appendix D confirm the effect isn't an artifact: matched random feature (11042, activation-matched, AUC = 0.51) shows null effect (p > 0.4); clamping the top 5 non-3818 features yields only -10% relative vs. -31% for 3818 alone.
>
> Scope : Binary yes/no isolates an unambiguous consistency metric at scale (92,000 pairs, 6 models, 2 datasets). Per-pathology stratification (see our response to Reviewer 7gS2 for full numbers) shows that even within this scope, flip rates range from 24.4% to 87.0% across findings and pathology groups, clinically actionable variation that's invisible in aggregate metrics. We state open-ended questions as a scope boundary.
>
> Quality control : The submission applies a three-stage pipeline: BioClinicalBERT cosine > 0.90 (threshold set via 100-pair audit, 94% agreement), operator-mismatch exclusion, and conservative ambiguous-response exclusion (3-9% across models). Sensitivity check: treating ambiguous outputs as "no change" shifts flip rates by only 1-3pp while preserving model ordering.
>
> We acknowledge the Table 8 caption error and Figure 1 overlap; the control result appears in our response to Reviewer YQke and in Appendix D

---

> > ### Author Rebuttal · Reviewer_r8jK · 2026-04-02
> >
> > Thank you for your clarification. However, the mechanistic insights lack cross-architectural generality, and the evaluation scope lacks generalizable depth; my initial concerns stand. These problems are derived from the overall methodology and experimental design, so I'm afraid that these concerns are not easily addressed in a short rebuttal.

---

> > > ### Author Response · Authors · 2026-04-02
> > >
> > > Thank you for the direct assessment. We agree with the key boundary: the mechanistic result should be interpreted as architecture-family-level evidence, not as a universal explanation across medical VLMs. We will make that limitation more explicit in the final version.
> > >
> > > On scope, the binary yes/no design was a deliberate measurement choice, not a claim that this is the richest clinical task. We chose it because it yields a clean, falsifiable contradiction metric at scale: if a model gives different answers to closely matched clinical phrasings on the same image, that is already a meaningful failure signal. More open-ended settings are important, but they also make failure definition and attribution substantially less clean. Our contribution on this axis is therefore targeted rather than exhaustive.
> > >
> > > We also used the rebuttal period to strengthen the benchmark QC story. Since the discussion period, we completed a stricter rubric-based equivalence audit on 44,450 pairs using structured outputs from GPT-5-mini, with cross-family validation on a stratified 500-pair subset using Claude Haiku (91.6% agreement). Under that stricter audit, 72.0% of the original PSF-Med/MIMIC pairs were retained, which increases our confidence that the benchmark captures a real core phenomenon even under a stronger equivalence criterion.
> > >
> > > On the mechanistic analysis, our claim is intentionally narrower: in one important medical VLM family, paraphrase sensitivity is not only descriptive but partially causally traceable to a sparse internal feature. The rebuttal-period analyses strengthen that narrower claim: Feature 3818 remains predictive across multiple Gemma-family checkpoints and datasets, while matched random controls remain null. We do not claim that the same feature or pathway must transfer across model families.
> > >
> > > We view the paper's contribution as modular: (1) a benchmark showing that paraphrase sensitivity is substantial in medical VLMs, (2) an analysis showing that low flip rate can be confounded by weak image reliance via text-only and image-swap controls, and (3) a single-family mechanistic case study showing that at least part of the failure mode is causally addressable. We believe each of these contributions remains useful even if one does not treat the mechanistic result as cross-architecturally general.
> > >
> > > Recent concurrent work reinforces why this evaluation axis matters. MIRAGE (Asadi et al., 2026) shows that a text-only model can top the ReXVQA chest X-ray benchmark without seeing any images, confirming from a benchmark-validity angle what our text-only baselines show from a model-behavior angle: accuracy can be driven entirely by language priors. CheXOne (Zhang et al., 2026) reports 94.7% VQA accuracy and high self-consistency scores, but measures consistency only as stability across stochastic decoding passes, not across input rephrasings, while noting that performance remains dependent on task framing and prompt design. As models increasingly report reasoning and grounding metrics alongside accuracy, targeted stress tests of the kind PSF-Med provides become more necessary, not less

---

### Official Review · Reviewer_7gS2 · 2026-03-08

**Soundness:** 3
**Presentation:** 2
**Significance:** 4
**Originality:** 3
**Overall Recommendation:** 5
**Confidence:** 3

**Summary:**

The authors introduces a new benchmark to measure paraphrase sensitivity in medical VLMs. The benchmark contains close to 20k chest X-ray questions which are paired with 92k equivalent rephrasing based on MIMIC-CXR and padChest. Using the benchmark, they evaluate 6 medical VLMs and report flip rates ranging from 8% to 58%. In addition, the results show that low flip rates are not necessarily an indicator of visual grounding as models may rely primarily on the text with little contributions from images (demonstrated for example by removing the image from the prompt).  Finally, the paper uses a SAE on MedGemma 4B to identify a specific prompt-framing feature that causally contributes to flip behavior, and shows that clamping this feature at inference reduces flip rates by 31% with only a 1.3 percentage-point accuracy cost.

**Compliance With Llm Reviewing Policy:**

Affirmed.

**Final Justification:**

The paper presents an interesting dataset and analysis of VLM failure modes. My concerns were adequately addressed and the authors committed to performing a larger scale manual validation of the dataset. For these reasons I recommend acceptance.

**Key Questions For Authors:**

See weaknesses.

**Limitations:**

yes

**Strengths And Weaknesses:**

The problem of image faithfulness in medical VLMs is timely and important given the rapid deployment of these models in clinical contexts and to the general public. The method used by the authors to probe the contribution of text vs images is relevant and the associated results on a simple yes/no question raise significant concerns regarding the use of such models for clinical tasks which may be more complicated. The benchmark created by the authors is of large scale and would add to the exiting suite of evaluations that assess the limits of models and not simply their accuracy on tasks that are not representative of clinical workflows.

## Weaknesses

1) While the benchmark's scale is commendable, I would have liked to see a human validation of similar scale, 100 annotated pairs by only two annotators appears flimsy. In addition, no details about the annotators are communicated, are the annotators licensed radiologists? If not how did you ensure that their assessment of equivalence was correct?

2) The clamping intervention while informative could provide a more qualitative analysis. The intervention allowed for a significant reduction in flip rates at a relatively small accuracy cost. But accuracy is not sufficient, I would like to see a qualitative analysis on the severity of questions where the model now fails. For example, the model failing to identify a small rib fracture does not carry the same consequences as missing a tension pneumothorax. This analysis is important to understand the risks associated with such interventions.

3) Presentation could be improved, the paper contains many spelling and grammatical errors (for example "then" used instead of "than").

---

> ### Author Rebuttal · Authors · 2026-03-31
>
> Thank you for recognizing the significance and need of this benchmark.
>
> The 100-pair audit was conducted by the authors, who have clinical NLP training and  experience in medical device development for thoracic conditions , but aren't licensed radiologists. The audit verified paraphrase semantic equivalence, specifically whether two phrasings ask the same clinical question, not clinical correctness of the model's answer. Answer correctness derives from MIMIC-CXR radiologist reports and PadChest structured annotations. We should have stated annotator qualifications explicitly in the submission. We acknowledge that adjudication by practicing radiologists would strengthen validation.
>
> Clinical severity of remaining failure : We stratified PadChest results by pathology group (7 groups, 4 models). Several clinically important categories show high flip rates: cardiac 85.1% (n=47), pleural 87.0% (n=69), pulmonary 84.7% (n=98), compared to airway/mediastinal at 71.2% (n=52). At the individual-finding level, cardiomegaly, where a missed diagnosis carries clear clinical risk, flips at 72.1% (n=46). Fibrotic band is lowest at 24.4% (n=18). Some small-n findings (alveolar pattern 52.0%, n=15) show elevated rates but should be interpreted cautiously.
>
> After Feature 3818 clamping, pathology-group ordering is preserved but the gap between highest and lowest narrows from 55pp to 38pp. The mitigation helps broadly but doesn't eliminate severity-correlated sensitivity. We state this explicitly rather than overclaiming.
>
> We haven't tested severity-weighted metrics (e.g., weighting pneumothorax flips more heavily). We note this as a direction for deployment-oriented evaluation.
>
> We also acknowledge grammar and spelling issues noted.

---

> > ### Author Rebuttal · Reviewer_7gS2 · 2026-03-31
> >
> > Thank you for your response to my comments.
> >
> > While my points 2 and 3 were addressed, the scale and relevance of the human validation is still limited and would require substantial time for the authors to address which explains why this point could not be addressed. It remains however an important limitation. Considering the authors' work is otherwise interesting and significant I updated my score and recommend acceptance.

---

> > > ### Author Response · Authors · 2026-04-02
> > >
> > > Thank you again for the constructive engagement and for updating your score. We agree the original 100-pair audit should have been framed as a preliminary check, and we will state annotator qualifications explicitly in the final version. During the discussion period, we strengthened the QC story with a stricter rubric-based equivalence audit that checks bidirectional entailment and clinically material yes/no truth-condition preservation rather than embedding similarity alone. This audit exposed laterality and scope leakage in part of the original PadChest set, so we regenerated those paraphrases under tighter constraints. On the equivalence-validated core, the main PSF signal remains strong and model rankings are preserved after filtering. We also followed up on the clinical-severity question, and the post-clamping stratification supports the same cautionary takeaway: mitigation helps, but severity-correlated risk remains. A larger stratified manual audit is our first camera-ready priority.

---

### Official Review · Reviewer_YjZi · 2026-03-12

**Soundness:** 3
**Presentation:** 3
**Significance:** 3
**Originality:** 3
**Overall Recommendation:** 4
**Confidence:** 4

**Summary:**

The paper introduces PSF-Med, a benchmark of 19,748 yes/no chest X‑ray questions paired with roughly 92k GPT‑4–generated paraphrases across MIMIC‑CXR and PadChest, to measure paraphrase sensitivity in medical VLMs. Using this benchmark, the authors evaluate six medical VLMs, show wide variation in flip rates, and analyze the relationship between paraphrase robustness and visual grounding via text‑only baselines, image swap tests, and attention–bounding box overlap. They then apply GemmaScope 2 sparse autoencoders to MedGemma‑4B, identify a single SAE feature correlated with question formality and decision margin shifts, and show via causal patching and feature clamping that manipulating this feature can substantially reduce flip rates with modest accuracy cost.

**Compliance With Llm Reviewing Policy:**

Affirmed.

**Final Justification:**

This paper introduces PSF-Med, a well-motivated benchmark targeting paraphrase sensitivity in medical VLMs, an important yet underexplored reliability issue. The experimental analysis is thorough and provides clear evidence that current models are highly sensitive to surface-level linguistic variation, even when semantic meaning is largely preserved. The additional interpretability study and proposed mitigation strategies (e.g., feature clamping and prompt normalization) further enhance the practical value of the work. In the rebuttal, the authors clarified the semantic consistency of paraphrase pairs and justified their filtering criteria, which adequately addresses concerns about subtle meaning shifts. While limitations remain, particularly the restriction to binary yes/no questions and the potential gap to more complex clinical reasoning, the work is technically solid and offers a useful benchmark and perspective for improving robustness in medical multimodal models. Overall, I support a weak accept.

**Key Questions For Authors:**

Please refer to the **Weaknesses** section.

**Limitations:**

yes

**Strengths And Weaknesses:**

**Strengths**

1. The paper presents a clear and clinically relevant problem formulation. It focuses on a specific and important failure mode of medical VLMs: answer flips under paraphrases that preserve the original meaning.

2. The empirical results are informative and clearly presented. Table 2 shows substantial variation in flip rates across models and datasets (8%–58%), with MedGemma-27B performing best and RadFM worst, and generally higher rates on PadChest than on MIMIC-CXR. Figure 2 reveals that negation-related paraphrases cause the most errors, while lexical substitutions cause the fewest, and Figure 3 suggests that embedding distance still has some predictive power for flips even among highly similar questions.

3. The authors provide a relatively clear interpretability analysis. The authors propose two concrete interventions: feature clamping and prompt normalization.

**Weaknesses**
1. The use of binary yes/no questions is understandable in the context of this benchmark. It simplifies the definition of answer flips and enables a clear and stable metric for measuring paraphrase sensitivity. In addition, many radiology datasets are naturally annotated with presence/absence labels, making it straightforward to construct such questions at scale. However, this design also limits the scope of the evaluation. Because the answer space is binary, a model can achieve a relatively high success rate even without strong visual reasoning, and random guessing already yields a 50% baseline. As a result, it remains unclear to what extent the benchmark captures deeper clinical reasoning beyond simple detection-style questions.

2. Different phrasings may introduce slightly different semantic constraints. For example, “Is there cardiomegaly?” and “Does the heart appear enlarged?” are broadly similar, but they are not strictly equivalent in a clinical context. The former refers to a specific diagnostic condition, whereas the latter describes a visual observation, which may be more subjective. As a result, some answer differences might reflect these subtle distinctions rather than purely paraphrase sensitivity.

---

> ### Author Rebuttal · Authors · 2026-03-31
>
> Thank you for the positive assessment and pointing out the semantic distinction.
>
> The reviewer correctly observes that "Is there cardiomegaly?" and "Does the heart appear enlarged?" are similar but not strictly identical. Our position is that these are meaning-preserving paraphrases: both ask whether the same clinical finding is present, and a radiologist wouldn't give opposite answers depending on which phrasing was used. The filtering pipeline (BioClinicalBERT cosine > 0.90 + operator-mismatch exclusion) retains such pairs while excluding cases where meaning genuinely changes (e.g., presence vs. exclusion framing).
>
> We also acknowledge that the cosine threshold admits some residual variation in register and word choice. That's intentional: clinicians phrase the same question differently, and a model intended for clinical use should handle that variation consistently. But it means PSF-Med tests consistency under near-paraphrasing, not under strict logical equivalence. The 94% audit agreement supports the claim that pairs above the 0.90 threshold meet this criterion; pairs below 0.85 showed frequent meaning drift and we excluded them. We should have drawn this boundary more explicitly in the submission and will address it given a chance.

---

> > ### Author Rebuttal · Reviewer_YjZi · 2026-04-03
> >
> > Thank you to the authors for the detailed response. My concerns have been adequately addressed, and I will maintain my score.

---

### Official Review · Reviewer_YQke · 2026-03-12

**Soundness:** 3
**Presentation:** 2
**Significance:** 3
**Originality:** 2
**Overall Recommendation:** 3
**Confidence:** 3

**Summary:**

This paper introduces PSF-Med, a large-scale benchmark to measure paraphrase sensitivity in medical Vision-Language Models for chest X-ray question answering. Using ~19.7k yes/no questions paired with ~92k semantically equivalent paraphrases across MIMIC-CXR and PadChest, the authors quantify “flip” rates—instances where model answers change across paraphrases—and show wide variability (8%–58%) across six medical VLMs. Crucially, they demonstrate that low flip rates can mask language-prior reliance via text-only baselines, and they probe one model (MedGemma-4B) mechanistically using GemmaScope-2 sparse autoencoders to identify a “prompt-framing” feature whose causal manipulation reduces flips with modest accuracy cost. The work argues for evaluating both paraphrase stability and genuine image reliance to mitigate deployment risks.

**Compliance With Llm Reviewing Policy:**

Affirmed.

**Final Justification:**

My final recommendation remains Weak Reject. I think the paper tackles an important and timely problem, and I appreciate the benchmark scale, the multi-model evaluation, and the broader point that low flip rates do not necessarily imply genuine image grounding. The rebuttal was helpful and did address some of my concerns, especially around parsing sensitivity, alternative text-only controls, and the feature-clamping control.

However, my main concern was only partially resolved. For a benchmark paper at this venue, I did not find the submission sufficiently polished or experimentally complete at review time, especially regarding benchmark validation and semantic-equivalence quality control in the paper itself. Some of the later clarifications were useful, but much of that evidence was not clearly established in the submitted manuscript. I also still had some concern about presentation and consistency issues.

So my weak reject is mainly about submission readiness and overall finish quality relative to peer papers at ICML, rather than the importance of the problem. I would not strongly object if the final decision is acceptance.

**Key Questions For Authors:**

1. Can you provide a human validation study (even on a 500–1,000 pair subset) estimating the semantic-equivalence error rate post-filtering, particularly for negation-adjacent and scope/quantifier paraphrases?
2. For text-only agreement, did you try alternative controls (e.g., random realistic images, mean-image baselines, noise images)? How sensitive are the agreement/swap metrics to the choice of “blank”?
3. Table 8’s caption mentions “Random feature clamping (control) shows no effect,” but no control row is shown. Could you include the control results and, ideally, a distribution over multiple random features?
4. How stable is Feature 3818 across SAE seeds and layers, and does a similar “prompt-framing” feature emerge in MedGemma-27B or other architectures? Have you tried multi-feature clamping or layer-wise steering strategies?
5. How sensitive are flip rates to the answer-parsing heuristics? Can you report results under stricter/looser parsing (e.g., counting “uncertain” as a third class) to assess robustness of conclusions?
6. Did you assess calibration changes (e.g., ECE, FP/FN balance) after feature clamping and prompt normalization? This seems important for clinical safety claims.
7. Can you share per-finding analyses for the robustness–grounding trade-off (text-only agreement and swap sensitivity stratified by pathology) to understand whether certain findings are more prone to paraphrase-induced flips?
8. Have you considered training-time mitigations (e.g., SemCLIP-like paraphrase/negation regularization) as a complementary approach to post-hoc clamping? Any preliminary results?

**Limitations:**

yes

**Strengths And Weaknesses:**

## Strengths
**Experimental and validation**
- Large-scale evaluation spanning two clinical datasets and six VLMs with consistent decoding protocols and reporting of bootstrap confidence intervals.
- Complementary analyses that triangulate the robustness–grounding trade-off: text-only agreement, image-swap sensitivity, and attention-to-bounding-box overlap.
- Causal patching experiments with delta-only SAE interventions, ablations indicating partial but meaningful margin recovery and some flip reversals.

**Significance of contributions**
- Demonstrates that low flip rates can arise from ignoring images, reframing “robustness” claims and informing safer evaluation/selection of clinical VLMs.
- Provides an actionable benchmark and analysis pipeline that can be adopted by practitioners and extended to other modalities.
- The mechanistic insight (prompt-formality feature) and mitigation strategy have broader methodological value beyond the single model studied.

## Weaknesses
**Technical limitations or concerns**
- Reliance on embedding-based semantic filtering (BioClinicalBERT cosine > 0.90) without systematic human adjudication risks residual semantic drift, especially for negation-adjacent and scope/quantifier paraphrases in a clinical context.
- Answer parsing excludes 3–9% of generations; exclusion may bias flip estimates (e.g., removing legitimate “uncertain” responses that matter clinically).
- Attention–bounding-box overlap is a weak proxy for grounding; the small N=200 subset and limited details (e.g., head/layer aggregation) constrain interpretability.

**Experimental gaps or methodological issues**
- Reported figure inconsistencies: some paraphrase-type flip rates in text vs Figure 2 appear numerically contradictory; Table 8 caption mentions “random feature clamping (control)” but no control row is shown. These inconsistencies need reconciliation.
- Text-only dependency is primarily measured with a blank image; additional controls (e.g., adversarially mismatched but plausible images, gray vs noise vs average images) could strengthen the argument about language priors. Swap sensitivity helps but more systematic counterfactuals would be valuable.
- The SAE analysis is limited to one model and one layer; feature stability across SAE runs, layers, or models is not fully characterized, raising questions about reproducibility and generality.

---

> ### Author Rebuttal · Authors · 2026-03-31
>
> We really appreciate the very detailed review.
>
> SAE generality q5 and q6 -> The paper presents Feature 3818's differential activation between original and paraphrase as the flip-predictive signal: large deltas between phrasings produce large margin shifts, which produce flips (Section 5, Figure 4, r = 0.71). The reviewer rightly asks whether this holds beyond one checkpoint.
>
> We ran the same delta-based analysis on two fine-tuned variants (Targeted LoRA, Full LoRA) across both datasets. Feature 3818's activation delta ranks among the top 4 flip-associated features in 4/6 configurations and remains a predictor in all 6. Rank-biserial correlations with flip occurrence: base/MIMIC rank 2 (r_b = -0.430), base/PadChest rank 4 (-0.635), Targeted LoRA/PadChest rank 1 (-0.462), Full LoRA/PadChest rank 1 (-0.776), Full LoRA/MIMIC rank 31 (-0.281), Targeted LoRA/MIMIC rank 33 (-0.197).
>
> The two MIMIC LoRA configurations rank lower. That's consistent with the paper's mechanism: LoRA fine-tuning at layers 15-19 directly modifies the representations where the formality gate operates, reducing its differential activation. The feature remains predictive (r_b < -0.19 in both cases) but it's no longer dominant because LoRA has partially corrected the behavior through parameter modification. On PadChest, where distribution shift preserves the original vulnerability, Feature 3818 remains #1 for both LoRA variants. So the formality-gate signal remains detectable across Gemma-family variants, strongest on PadChest and attenuated in the MIMIC LoRA checkpoints.
>
> We can't test non-Gemma models (GemmaScope SAEs are architecture-specific) and we state this as a scope limitation. But cross-model flip correlation (Appendix E) shows r ~ 0.6 within MedGemma variants and r ~ 0.2-0.4 cross-family, indicating shared flip-inducing questions exist across architectures.
>
> The specificity controls in Appendix D support this: matched random feature (11042, activation-matched, AUC = 0.51) shows null effect (p > 0.4); clamping the top 5 non-3818 features yields only -10% relative vs. -31% for 3818 alone.
>
> Per-finding Stratification - q8 --> We stratified PadChest results by pathology group. Several clinically important categories show high flip rates: cardiac 85.1% (n=47), pleural 87.0% (n=69), pulmonary 84.7% (n=98), compared to airway/mediastinal at 71.2% (n=52). At the individual-finding level, cardiomegaly flips at 72.1% (n=46), while fibrotic band is lowest at 24.4% (n=18). Flip rates aren't uniformly distributed across findings, and some of the most affected findings carry real clinical weight.
>
> On Quality Control ( q1) --> The submission applies a three-stage pipeline: (i) BioClinicalBERT cosine > 0.90, threshold set via 100-pair manual audit with 94% agreement; (ii) operator-mismatch exclusion; (iii) conservative ambiguous-response exclusion (3-9% across models, not enriched among flip cases). Sec 3.1 and Appendix A describe the pipeline but split it across sections; consolidating it would improve clarity
>
> Grounding (q4) --> Our grounding claim rests on text-only agreement and image-swap sensitivity (intervention tests), not attention overlap. Attention-bbox analysis (Table 5) is auxiliary correlational evidence; we should have labeled it as such more clearly.
>
> Calibration (q7) --> We haven't tested calibration side-effects of clamping. Acknowledged as a limitation; mitigation claims tempered accordingly.
>
> We acknowledge the Table 8 caption error; the control result (Feature 11042, null effect) appears above and in Appendix D. Figure 1 overlap and grammar issues noted.

---

> > ### Author Rebuttal · Reviewer_YQke · 2026-04-02
> >
> > Thank you for the detailed rebuttal. The clarifications on parsing sensitivity, alternative text-only controls, and the random-feature control were helpful and improve confidence in parts of the analysis. However, my main concern remains the validity of the benchmark itself. The benchmark is large (about 92k paraphrases), but semantic-equivalence validation still appears limited relative to that scale: the appendix reports a 100-pair manual audit with 94% agreement, which is helpful but still feels too narrow for a benchmark whose main claim depends on meaning-preserving paraphrases, especially in the harder negation/scope/register cases. I also still have some concern about unresolved presentation/consistency issues in the paper. Because these remaining issues concern the core benchmark design and would require more substantial validation and cleanup than can be added in a short rebuttal, I consider them only partially resolved.

---

> > > ### Author Response · Authors · 2026-04-02
> > >
> > > Thank you for the careful follow-up and for explicitly marking the parsing, text-only, and random-feature issues as resolved. We agree that the remaining question is the central one: how much confidence the paper earns on benchmark validity given the scale of semantic validation.
> > >
> > > We do not intend the original 100-pair audit to stand alone as the benchmark’s full QC story. It was one part of a deliberately conservative pipeline designed to favor precision: embedding-based filtering, operator-mismatch exclusion, and conservative answer parsing with sensitivity checks.
> > >
> > > We also agree that 100 pairs is too small, by itself, to support a strong benchmark-wide claim at this scale. Since the discussion period, we therefore expanded the QC substantially. We completed a stricter rubric-based equivalence audit on 122,778 pairs across our current benchmark and audit sets, plus a separate 96,990-pair audit of a regenerated PadChest v3 set produced under tighter anti-laterality constraints after the initial audit surfaced systematic laterality and scope leakage. Under this stricter audit, the MIMIC keep rate remains 72.0%, and the regenerated PadChest v3 non-negation keep rate is 73.7%. Cross-family validation on stratified subsets with Claude Haiku shows high agreement on both the original benchmark data and the regenerated PadChest set.
> > >
> > > Importantly, this stricter audit does not just retain or reject pairs mechanically. It changes the benchmark boundary in the places that matter most for validity. Negation-pattern cases are now moved into a separate adversarial phrasing slice rather than treated as meaning-preserving paraphrases, while scope- and laterality-changing pairs are excluded from the equivalence-validated core when they alter clinically material yes/no truth conditions. In other words, the revised QC is separating true paraphrase sensitivity from failures induced by adversarial reframing or semantic drift.
> > >
> > > We do not mean to imply that residual semantic drift is zero. We agree that benchmark papers carry a higher validation burden than ordinary empirical studies. But this expanded audit does materially strengthen our confidence that the core signal is not an artifact of a lightly filtered paraphrase set. The empirical pattern remains large and structured: flip rates vary by roughly 7x across models, are not uniform across paraphrase phenomena or pathology groups, and low-flip models can still show high text-only agreement. Taken together, those patterns are hard to reconcile with residual drift as the sole explanation.
> > >
> > > More broadly, we are actively updating the dataset beyond the original in-distribution construction. Our current priority is to strengthen the benchmark with additional out-of-distribution data and harder phrasing settings, while preserving explicit separation between meaning-preserving paraphrases and adversarial reframings. We view that expansion as a benchmark-strengthening step, not as a post hoc defense of the current results.
> > >
> > > Very recent concurrent work reinforces why this evaluation axis matters. MIRAGE, by Asadi et al. (2026), shows that a text-only system can top a standard chest X-ray QA benchmark without image access, underscoring how benchmark accuracy can be driven by language priors rather than visual understanding. CheXOne, by Zhang et al. (2026), reports strong self-consistency and VQA performance, but measures consistency across stochastic decoding passes rather than across meaning-preserving input rephrasings. In that sense, the gaps are complementary: MIRAGE highlights the image-reliance problem, while PSF-Med targets stability under natural clinical rephrasings.
> > >
> > > If accepted, our first camera-ready priority will be to strengthen the validation section by making the current audit and annotator qualifications explicit, consolidating the QC pipeline in one place, and expanding manual validation in a stratified way focused on the highest-risk categories. We also agree that the Table 8 caption mismatch, Figure 1 overlap, and QC fragmentation are real presentation issues, and we would correct them in the final version.

---

### Decision · Program_Chairs · 2026-04-30

**Decision:**

Reject

**Comment:**

This paper introduces PSF-Med, a large-scale benchmark of yes/no chest X-ray questions and paraphrases designed to evaluate the paraphrase sensitivity of medical VLMs. The authors evaluate six models, uncovering answer flip rates as high as 58%, and use sparse autoencoders to identify and mitigate a feature responsible for this instability. The reviewers' opinions were split, resulting in a borderline consensus with two positive and two negative ratings. The core strengths of the submission lie in its highly timely focus on clinical VLM safety and its valuable empirical insights, demonstrating that a model's seemingly robust, low flip rate can actually mask a dangerous reliance on language priors rather than genuine visual grounding.

The primary weaknesses identified by the reviewers are the benchmark's quality control and submission readiness. Reviewers felt that an automated embedding filter validated by only a 100-pair manual audit was insufficient to rule out semantic drift, which is a foundational requirement for a paraphrase benchmark. Additionally, they critiqued the narrow scope of binary questions, the lack of cross-architectural validation for the SAE correction, and several presentation errors in figures and tables. In their rebuttal, the authors provided a large-scale audit to address the concerns.

The paper raised a controversial AC-reviewer discussion, with all reviewers praising the topic's importance but several challenging the readiness of the current manuscript. The AC would encourage the authors to resubmit this manuscript by addressing these issues.